# Combined Styletubation with Videolaryngoscopy for Tracheal Intubation in Patients Undergoing Thyroidectomy with Intraoperative Neuromonitoring

Hui-Shan Pan [1], Tiffany Corey [2], Hsiang-Ning Luk [1,*], Jason Zhensheng Qu [3] and Alan Shikani [4,5]

1 Department of Anesthesia, Hualien Tzu-Chi Hospital, Hualien 970, Taiwan; apan@tzuchi.com.tw
2 CRNA Class of 2023, College of Nursing, Rush University, Chicago, IL 60612, USA; tiffany_c_corey@rush.edu
3 Department of Anesthesia, Critical Care and Pain Medicine, Massachusetts General Hospital, Harvard Medical School, Boston, MA 02115, USA; jqu@mgh.harvard.edu
4 Division of Otolaryngology—Head and Neck Surgery, LifeBridge Sinai Hospital, Baltimore, MD 21040, USA; ashikani@gmail.com
5 Division of Otolaryngology—Head and Neck Surgery, MedStar Union Memorial Hospital, Baltimore, MD 21218, USA
* Correspondence: lukairforce@tzuchi.com.tw or lukairforce@gmail.com

**Abstract:** The purpose of this case series report is to demonstrate the current state of the art regarding tracheal intubation of an evoked electromyography-endotracheal tube (EMG-ET tube) for continuous intraoperative recurrent laryngeal nerve monitoring (IONM) in patients undergoing thyroid surgery. Both direct laryngoscopy (DL) and videolaryngoscopy (VL) are popular for routine tracheal intubation of an EMG-ET tube. A new intubating technique (styletubation), using a video-assisted intubating stylet (VS), provides less traumatic and swift intubation. Styletubation combined with VL ensures the precise placement of the EMG-ET tube. This novel intubation technique improves the outcome of intubating an EMG-ET tube for IONM.

**Keywords:** styletubation; video-assisted intubating stylet; tracheal intubation; laryngoscope; videolaryngoscope; anesthesia; difficult airway; thyroidectomy; parathyroidectomy; intraoperative neuromonitoring; recurrent laryngeal nerve; vocal cord paralysis





## 1. Introduction

During a thyroidectomy, visualizing and identifying the recurrent laryngeal nerve (RLN) are crucial and helpful to prevent complications of nerve injury and reduce the sequelae of postoperative vocal cord paralysis. For such a purpose, the introduction of intraoperative nerve monitoring (IONM) enhances the identification of the RLN by providing functional dynamics of evoked electromyography (evoked EMG) elicited by stimulating the specific nerve. Such intraoperative monitoring therefore has been used as an adjunct to assist in identifying the RLN and predicting function after dissection [1–5]. Whether IONM should be routinely applied or under specific conditions (e.g., second operations, total thyroidectomy with neck dissection, and cases of malignant diseases), one thing for sure is that IONM does make thyroid surgery safer. Interestingly, it was reported that the use of IONM during thyroid surgery was not associated with the risk for postoperative vocal cord paralysis [6] or RLN palsy [7].

Difficult or failed intubation is a major contributor to morbidity for patients and liability for anesthesiologists [8]. The overall prevalence of difficult intubation (DI) varies widely from 0.1% to 10.1% [9], 8% to 37% [10], 5.8% [11], 2.4% to 2.7% [12], and 0.16% [13]. Such variation in the incidence of DI comes from various reasons, such as the different definitions applied, patient's factor, types of surgery, clinical settings and locations, severity of critical clinical scenarios, equipment and organizational structures, and airway operator's experiences and skills. One of the contributing factors is the patient's anatomy over the head

and neck region. For example, while conducting general anesthesia in patients undergoing a thyroidectomy, difficult tracheal intubation might occur due to the mass effect of the goiter or tumor. The overall incidence of difficult intubation in a thyroidectomy varied, e.g., 5.3% [14], 5.5% [15], 7% [16], 8.5% [17], 9.6% [18], and 11.1% [19].

In patients undergoing a thyroidectomy, conventional direct laryngoscopes (DL) have routinely been used [20]. When the airway is compromised by the size, nature, and localization of the large thyroid glands or tumors, the tracheal intubation can be very challenging to the airway operators. The alternative airway management modalities then include DL [21–23], videolaryngoscopes (VL) [21,23–25], optic intubating stylets [20], awake fiber-optic bronchoscopes (FOB) [23,26–28], combined FOB with VL [29,30], awake tracheostomy [31], and the use of extracorporeal circulation membrane oxygenation (ECMO) prior to tracheal intubation [32–35].

Quite often, the application of IONM during a thyroidectomy is patient-centered or surgeon-centered. When that happens, the airway operator would face other airway technical considerations related to the electromyographic endotracheal tube (EMG-ET tube) issues. Namely, the placement and position of such an EMG-ET tube may be difficult and need to be checked, adjusted, and optimized. Such a need can be met by using either DL/FOB [36–39], VL [36,40–43], and an optic intubating stylet [36,44,45]. In this brief report, we demonstrate the routine application of combined styletubation with videolaryngoscopy for tracheal intubation in patients undergoing a thyroidectomy with intraoperative IONM. The styletubation technique using a video-assisted intubating stylet (VS) for tracheal intubation has recently been reviewed and reported [46–48]. In patients undergoing a thyroidectomy, we applied the combined styletubation with videolaryngoscopy to place the EMG-ET tube for IONM (Figure 1). Briefly, after anesthesia was induced, we applied with a VL to expose the patient's glottis (Figure 1A,D). Then, a VS was utilized to intubate an EMG-ET tube into the patient's trachea (Figure 1B,E). Finally, the position was verified with the VL again. In our operating room setting, we routinely utilized (1) the C-MAC-VS (video stylet), KARL STORZ SE & Co. KG, Tuttlingen, Germany; (2) the TuoRen Kingtaek video intubating stylet, TuoRen, Henan TuoRen Medical Device Co., Xinxiang 453401, China; (3) the UE rigid laryngoscope and TRS video stylet, Zhejiang UE Medical Corp., Taizhou 317300, China; and (4) the Trachway video-intubating stylet, Markstein Sichtech Medical Corp., Taichung 407, Taiwan.

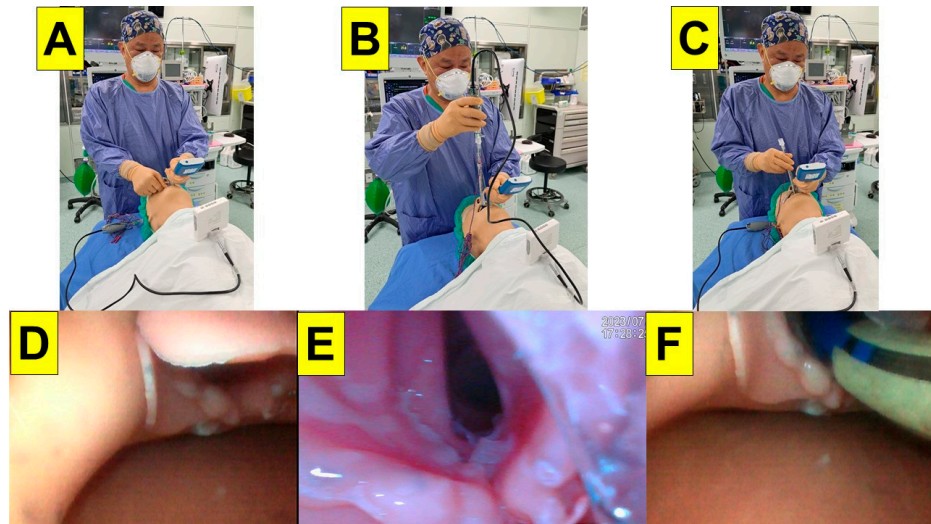

**Figure 1.** Demonstration of combined styletubation with videolaryngoscopy to perform tracheal intubation with an EMG-ET tube for IONM in a mannequin model. (**A**,**D**): Laryngoscopy and its view on a video-monitor (UE rigid videolaryngoscope). (**B**,**E**): With the aid of VL, tracheal intubation of an EMG-ET tube was achieved with styletubation (S-RVL video stylet, Sensorendo Medical Technology, Zhuhai, China). (**C**,**F**): After withdrawing the video stylet (VS), the proper placement and position were finally verified by the VL.2. Case presentation.

Conventionally, tracheal intubation for a thyroidectomy was performed with an endotracheal tube without intraoperative neuromonitoring. In Figures 2 and 3, a 44-year-old woman with a BMI of 33.6 kg/m$^2$ (height 157 cm and weight 83 kg) underwent right total thyroidectomy due to a goiter (3 × 6 × 7 (cm) in size), as an example. Tracheal intubation was performed with a conventional endotracheal tube without IONM. The combined VL with styletubation technique was applied in this patient whose pre-operative airway evaluation parameters were a mouth-opening of 3.5 cm, thyromental distance of 8 cm, sternomental distance of 16 cm, neck circumference of 42 cm, modified Mallampati class III, and upper lip bite test class 1. The intubation procedure was smooth (styletubation: from lip to trachea, 6 s). The Cormack–Lehane grading under videolaryngoscopic view was Figure 2B and the percentage of glottic opening (POGO) score under styletubation was 80% (Figure 3).

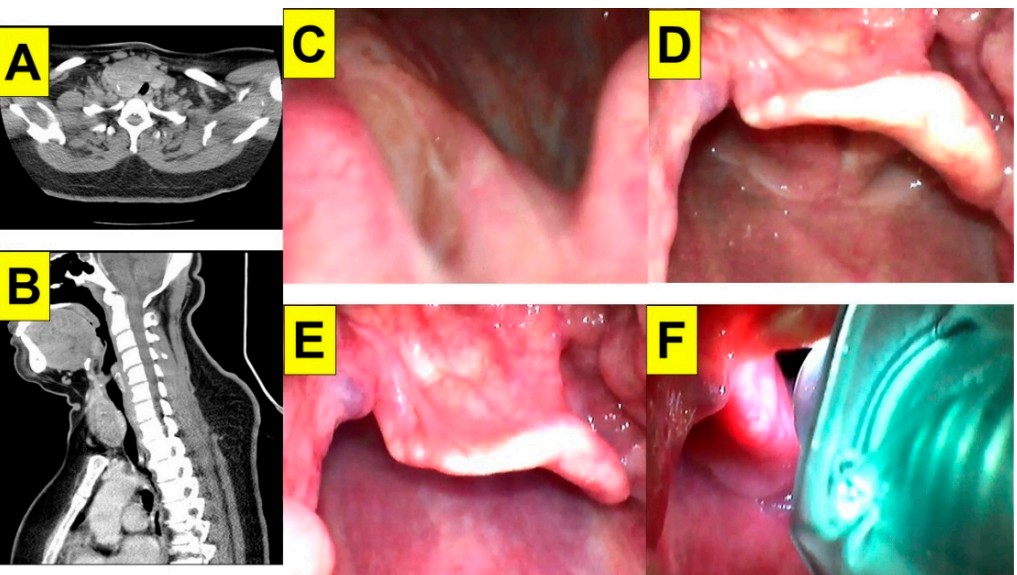

**Figure 2.** Combined styletubation with VL in a female patient (44 year old, BMI 33.6 kg/m$^2$) undergoing thyroidectomy for thyroid carcinoma. (**A,B**): Views of computed tomography; (**C–F**): views from the VL; (**C**): view of uvula; (**D**): glottic view with lifting the blade; (**E**): without lifting the blade; (**F**): visualization of advancement of the standard ET tube into trachea. (See Video S16 in the Supplementary Materials).

In this case series report, we describe 15 patients who presented to our medical center for a routine thyroidectomy (Table 1). All these patients received an IONM EMG-ET tube (Medtonic NIM TRIVANTAGE$^{TM}$ (Medtronic, 710 Medtronic Parkway Minneapolis, MN 55432-5640, USA) EMG endotracheal tube, I.D. 6 mm or 7 mm; or Medtronic NIM™ EMG endotracheal tube) for continuous intraoperative neuromonitoring. Pre-operatively, a full medical history and physical exam (including airway examination) were obtained from each patient. All the patients were classified by the American Society of Anesthesiologists (ASA) physical status. The anesthesia plan was general anesthesia with endotracheal intubation using an EMG-ET tube. All the standard ASA monitoring (including ECG; NIBP; SpO$_2$; ToF; and ETCO$_2$) was applied intraoperatively. The induction and maintenance of anesthesia followed the routine protocols, including medications of midazolam; fentanyl; propofol; rocuronium; succinylcholine; and sevoflurane or desflurane. Other relevant information related to the patients is summarized in Table 1. All the tracheal intubations were performed by the styletubation with a VS, in combination with a VL (as demonstrated in Figure 1). All the intubation processes were smooth, accurate, and swift (Table 1).

**Table 1.** A summary of the 15 cases in this series report.

| | Case 1 | Case 2 | Case 3 | Case 4 | Case 5 | Case 6 | Case 7 | Case 8 | Case 9 | Case 10 |
|---|---|---|---|---|---|---|---|---|---|---|
| Age/Gender | 64/F | 71/F | 45/F | 33/F | 54/F | 70/F | 41/F | 80/F | 54/F | 54/F |
| Height (cm)/weight (kg) | 153/64 | 154/52 | 156/60 | 148/62 | 153/53 | 147/47 | 151/55 | 152/55 | 161/72 | 145/56 |
| BMI (kg/m$^2$) | 27.3 | 21.9 | 24.6 | 28.3 | 22.6 | 21.7 | 24.1 | 23.8 | 27.7 | 26.6 |
| ASA physical status | II | II | I | I | I | II | I | III | III | II |
| Comorbidity | Myocardial ischemia (T-wave inversion) | Hypertension | Nil | Nil | Hashimoto's thyroiditis, multinodular goiter with tracheal compression and deviation | Hypertension | Nil | Hypertension, chronic kidney disease | Hypertension, cerebral aneurysm | Diabetes |
| Diagnosis | Bilateral mutinodular goiter | Parkinson's disease | Thyroid cyst, left | Thyroid papillary carcinoma, left | Papillary microcarcinoma of thyroid | Bilateral multinodular goiter with tracheal deviation | Left thyroid goiter | Bilateral multiple nodular goiter, 3 cm lesion of left thyroid lower pole, T | Thyroid nodular goiter | Bilateral multinodular goiter |
| Major surgery | Bilateral total thyroidectomy; left upper parathyroid gland autotransplantation | Total thyroidectomy | Excision of thyroid cyst, left | Total thyroidectomy | Total thyroidectomy | Total thyroidectomy | Left total thyroidectomy | Total thyroidectomy | Lobectomy (right), partial thyroidectomy (left) | Total thyroidectomy |

**Table 1.** *Cont.*

| | Case 1 | Case 2 | Case 3 | Case 4 | Case 5 | Case 6 | Case 7 | Case 8 | Case 9 | Case 10 |
|---|---|---|---|---|---|---|---|---|---|---|
| Size of resected thyroid mass | 0.8 to 2.7 cm | 70 to 94 gm | 2.8 × 1.3 (cm) | 1.4 × 0.9 (cm) | (Right lobe): 7.0 × 5.0 × 3.8, (left lobe): 4.8 × 4.2 × 2.3 (cm); (tumor size: 0.2 cm). | (Right lobe) 6.2 × 5.5 × 4.0; (left lobe) 5.5 × 4.0 × 2.0 (cm) | 3.5 × 3.0 × 1.8 (cm) nodular goiter with hemorrhagic cyst | (Right lobe) 5.5 × 3.0 × 2.2 cm; (left lobe) 6.0 × 4.5 × 3.6 (cm); (tumor) (right 1 cm) and (left 0.4 cm) | 0.74 × 0.8 (cm) | 6.8 × 5.0 × 3.6 (cm) |
| Induction | MDZ, FEN, PPF, ROC (0.3 mg/kg), SCh (1.4 mg/kg) | MDZ, FEN, PPF, ROC (1.0 mg/kg) | MDZ, FEN, PPF, ROC (0.3 mg/kg), SCh (1.4 mg/kg) | MDZ, FEN, PPF, ROC (0.6 mg/kg), SCh (1.3 mg/kg) | MDZ, FEN, PPF, ROC (1 mg/kg) | MDZ, FEN, PPF, ROC (0.4 mg/kg), SCh (2 mg/kg) | MDZ, FEN, PPF, ROC (0.2 mg/kg), SCh (2 mg/kg) | MDZ, FEN, PPF, ROC (0.8 mg/kg) | MDZ, FEN, PPF, ROC (0.7 mg/kg) | MDZ, FEN, PPF, ROC (0.9 mg/kg) |
| Maintenance | Sevoflurane | Desflurane | Sevoflurane | Sevoflurane | Sevoflurane | Sevoflurane | Sevoflurane | Sevoflurane | Sevoflurane | Sevoflurane |
| MMT | III | II | III | II | III | II | II | III | II | III |
| ULBT | 1 | Class 1 | Class 2 | Class 1 | Class 1 | Class 1 | Class 1 | Class 2 | Class 1 | Class 2 |
| Inter-incisor distance | 4 cm | 4 cm | 3.5 cm | 4 cm | 4 cm | 4 cm | 4 cm | 5 cm | 4 cm | 4.5 cm |
| Sternomental distance | 15 cm | 13 cm | 17 cm | 16 cm | 16 cm | 16 cm | 16 cm | 15 cm | 15 cm | 17 cm |
| Neck circumference | 34 cm | 39 cm | 31 cm | 34 cm | 34 cm | 35 cm | 38 cm | 39 cm | 36 cm | 34 cm |
| C-L grading | 2b | 2a | 2b | 2a | 2a | 2a | 2b | 2b | 1 | 3 |
| LQS grading | Grade 1 | Grade 1 | Grade 1 | Grade 1 | Grade 2 | Grade 1 | Grade 1 | Grade 2 | Grade 1 | Grade 2 |
| View of glottis (POGO grading) | 100% | 100% | 100% | 100% | 90% | 80% | 90% | 100% | 100 % | 60% |
| Success on first attempt | Yes | Yes | Yes | Yes | Yes | Yes | Yes | Yes | Yes | Yes |
| Intubation time (VS) | 9 s | 4 s | 33 s | 15 s | 14 s | 8 s | 9 s | 7 s | 5 s | 10 s |
| V1 verification | Yes | Yes | Yes | Yes | Yes | Yes | Yes | Yes | Yes | Yes |

**Table 1.** *Cont.*

| | Case 1 | Case 2 | Case 3 | Case 4 | Case 5 | Case 6 | Case 7 | Case 8 | Case 9 | Case 10 |
|---|---|---|---|---|---|---|---|---|---|---|
| Complications | Nil | Nil | Nil | Nil | Nil | Nil | Nil | Nil | Nil | Nil |
| Subjective satisfaction | Easy and excellent | Easy and excellent | Awkward | Easy and excellent | Easy and excellent | Easy and excellent | East and excellent | Easy and excellent | Easy and excellent | Easy and excellent |
| Supplementary materials | Video S1 | Video S2 | Video S3 | Video S4 | Video S5 | Video S6 | Video S7 | Video S8 | Video S9 | Video S10 |

| | Case 11 | Case 12 | Case 13 | Case 14 | Case 15 |
|---|---|---|---|---|---|
| Age/gender | 71/F | 55/F | 21F | 25/F | 51/F |
| Height (cm)/weight (kg) | 159/53 | 157/72 | 157/63 | 164/64 | 152/73 |
| BMI (kg/m$^2$) | 20.9 | 29.2 | 25.5 | 25.5 | 31.5 |
| ASA physical status | II | I | I | I | I |
| Comorbidity | Hypertension | Nil | Nil | Nil | Nil |
| Diagnosis | Multinodular goiter (left) | Thyroid nodule (right) | Papillary carcinoma, right thyroid; multiple nodular goiter, left | Papillary thyroid carcinoma, right | Bilateral thyroid cysts |
| Major surgery | Hemithyroidectomy (left) | Right hemithyroidectomy, left lobectomy | Total tyroidectomy, neck lymph node dissection | Total thyroidectomy | Bilateral subtotal thyroidectomy |
| Size of resected thyroid mass | 0.8 to 2.7 cm | $1.4 \times 1.2 \times 1.2$ (cm) | $1.78 \times 1.3$ (cm) | 1.8 cm in length | (Right lobe): $3.0 \times 3.2$; (left lobe): $2.5 \times 3.0$ (cm) |
| Induction | MDZ, FEN, PPF, ROC (0.6 mg/kg), SCh (1.5 mg/kg) | MDZ, FEN, PPF, ROC (0.6 mg/kg) | MDZ, FEN, PPF, ROC (0.8 mg/kg) | MDZ, FEN, PPF, ROC (0.6 mg/kg) | MDZ, FEN, PPF, ROC (0.7 mg/kg) |
| Maintenance | Sevoflurane | Sevoflurane | Sevoflurane | Sevoflurane | Sevoflurane |
| MMT | III | II | II | II | II |
| ULBT | Class 3 | Class 1 | Class 1 | Class 1 | Class 1 |

**Table 1.** *Cont.*

| | Case 11 | Case 12 | Case 13 | Case 14 | Case 15 |
|---|---|---|---|---|---|
| Inter-incisor distance | 3.5 cm | 3.5 cm | 5.5 cm | 4 cm | 3.5 cm |
| Sternomental distance | 15 cm | 14 | 15 cm | 15 cm | 19 cm |
| Neck circumference | 31 cm | 38 | 35 cm | 34 cm | 40 cm |
| C-L grading | 1 | 3 | 2a | 1 | 2a |
| LQS grading | Grade 1 | Grade 2 | Grade 1 | Grade 1 | Grade 1 |
| View of glottis (POGO grading) | 90% | 100% | 100% | 100% | 100% |
| Success on first attempt | Yes | Yes | Yes | Yes | Yes |
| Intubation time (VS) | 6 s | 10 s | 11 s | 10 s | 6 s |
| V1 verification | Yes | Yes | Yes | Yes | Yes |
| Complications | Nil | Nil | Nil | Nil | Nil |
| Subjective satisfaction | Easy and excellent | Easy and excellent | Easy and excellent | Easy and excellent | Easy and excellent |
| Supplementary materials | Video S11 | Video S12 | Video S13 | Video S14 | Video S15 |

F/M: female/male; BMI: body mass index; ASA: American Society of Anesthesiologists; MDZ: midazolam; FEN: fentanyl; PPF: propofol; SCh: succinylcholine; ROC: rocuronium; MMT: modified Mallampati test; ULBT: upper lip bite test; C-L grading: Cormack–Lehane grading; VL: videolaryngoscopy; VS: video-assisted intubating stylet technique (styletubation); V1: vagus nerve EMG signal; LQS grading: glottis visualization using styletubation (Grade 1—whole part of vocal cords visible; Grade 2—minimal space between epiglottis and posterior pharyngeal wall; Grade 3—no space between epiglottis and posterior pharyngeal wall).

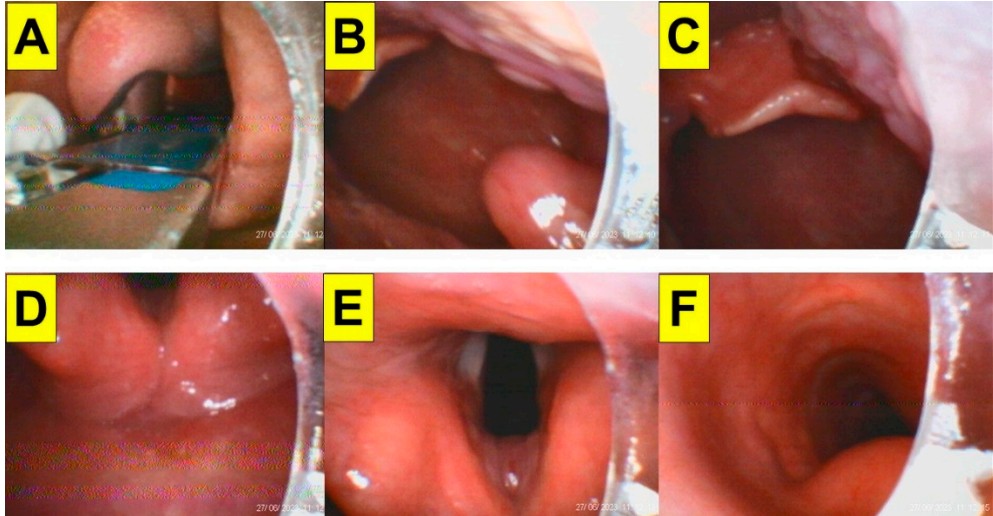

**Figure 3.** Combined styletubation with VL in the same patient in Figure 2. Views taken from the VS. (**A**): View of the blade inside the mouth before inserting the intubating stylet; (**B**): views of uvula and epiglottis; (**C**): view of epiglottis without lifting by the blade; (**D**,**E**): visualization of the glottic area; (**F**): view of the tracheal rings. (See Video S17 in the Supplementary Materials).

The proper position of the EMG-ET tube in the larynx is crucial to the functionality of IONM in thyroid surgery. In order to acquire high-quality signals from IONM, the electrodes embedded along the surface of the EMG-ET tube must be positioned in good contact with the vocal cords. A malpositioned or improper size of the EMG-ET tube may cause a false decrease in or loss of the EMG signals. For example, false signals may make it difficult to distinguish a true RLN injury and therefore lead to a wrong surgical decision. In all 15 cases, we routinely applied a VL to facilitate the intubation and at the same moment to verify the optimal placement of an EMG-ET tube (Table 1). The incidence of difficult tracheal intubation (e.g., measured by the Cormack–Lehane grading by laryngoscopy or the intubation difficulty scale) varied, depending on many factors. For tracheal intubation with an EMG-ET tube in patients undergoing a thyroidectomy, we did not encounter such difficulty using the styletubation technique (Table 1). Figure 4 shows a patient (Case 1 in Table 1) with Cormack–Lehane grading 2b (Figure 4A) under laryngoscopy. The EMG-ET tube was placed with styletubation which manifested an LQS grade 1 (i.e., any part of the glottis can be visualized; Figure 4D) with a POGO score of 100% (Figure 4E). The intubation (time to trachea) was 9 s. Meanwhile, the proper placement of the EMG-ET tube was visually verified with the videolaryngoscope (Figure 4C).

Similar to higher grades of the Cormack–Lehane classification by laryngoscopy, a small proportion of the patients showed restricted/limited glottic visualization by styletubation. Such a case was presented in Figure 5 (Case 12 in Table 1). The Cormack–Lehane grading 3 (Figure 5A) is shown under laryngoscopy. The EMG-ET tube was placed with styletubation which manifested an LQS grade 2 (i.e., no part of the glottis can be visualized, but there is still some space between the epiglottis and posterior pharyngeal wall; Figure 5D) with a POGO score of 100% (Figure 5E). The intubation (time to trachea) was 10 s.

Although the styletubation technique is the main intubating modality in our medical center, whether the combined styletubation with VL is easy to handle by junior residents needs to be observed. Figure 6 shows the combined technique performed by a junior airway operator in a 45-year-old woman receiving tracheal intubation with an EMG-ET tube (Case 3 in Table 1). The intubating process took a little bit longer (33 s) than usual.

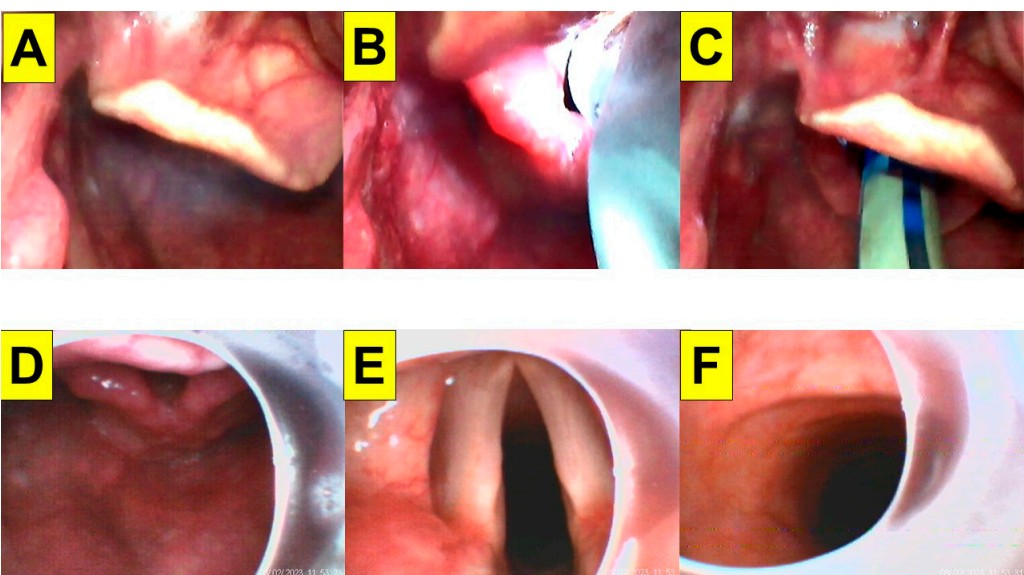

**Figure 4.** (Case 1 in Table 1). Serial images recorded during laryngoscopy and styletubation. (**A**–**C**): Views acquired by laryngoscopy. (**D**–**F**): Views obtained by styletubation. C-L grade was 2b and LQS grade was 1 (see Video S1 in the Supplementary Materials).

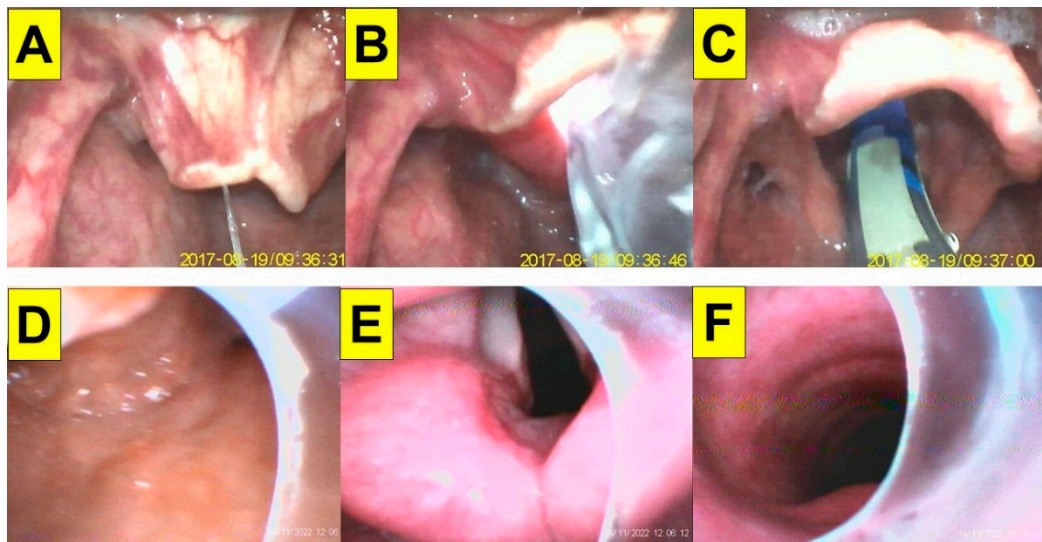

**Figure 5.** (Case 12 in Table 1). Serial images recorded during laryngoscopy and styletubation. (**A**–**C**): Views acquired by laryngoscopy. (**D**–**F**): Views obtained by styletubation. (See Video S12 in the Supplementary Materials).

The patients undergoing thyroid surgery are commonly placed in a hyperextension position for better exposure of the surgical field (Figure 7A–C). When the patient's neck was extended by a roll under their shoulders, the depth and position of the EMG-ET tube may be therefore changed or displaced, and subsequently caused a malfunction of the IONM. Therefore, a proper position of the EMG-ET tube could be assured and achieved by a modified protocol. After the induction of anesthesia, the patient was first placed in a hyperextended position (the Rose position, Figure 7A–C). Then, the EMG-ET tube was placed by the combined VL (Figure 7D–F) and styletubation (Figure 8). The final verification of the proper length and position of the EMG-ET tube was performed with the aid of the VL (Figure 7E,F).

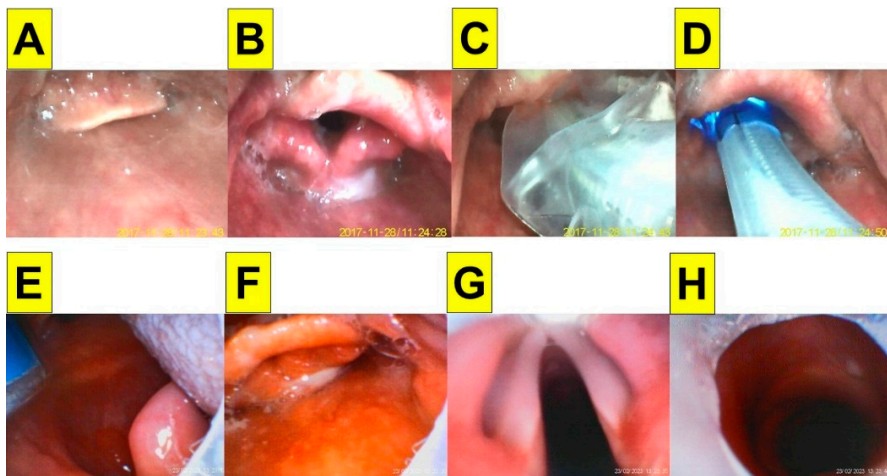

**Figure 6.** (Case 3 in Table 1). Serial images recorded during laryngoscopy and styletubation performed by a junior airway operator. (**A–D**): Views acquired by laryngoscopy. (**E–H**): Views obtained by styletubation. (See Video S3 in the Supplementary Materials).

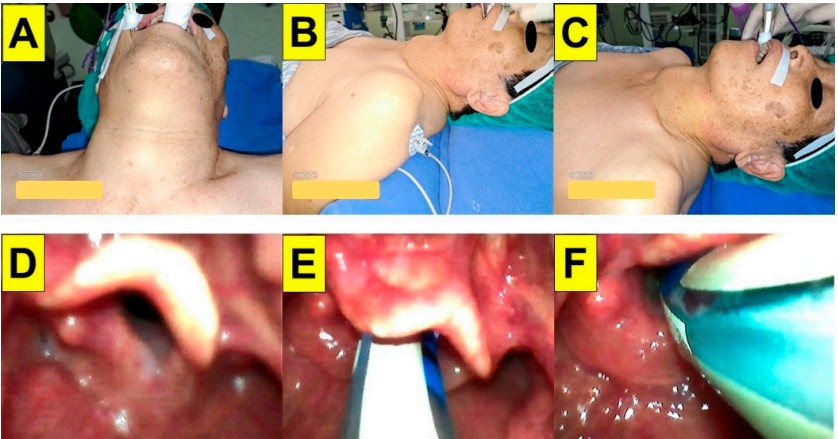

**Figure 7.** (Case 2 in Table 1). Serial images recorded during laryngoscopy. The Rose position was achieved with the aid of a shoulder roll under the patient before intubation. (**A–D**): Views acquired by laryngoscopy. (**E,F**): Views from a VL after placement of an EMT-ET tube by styletubation. A proper position and length of the tube was subsequently verified by VL.

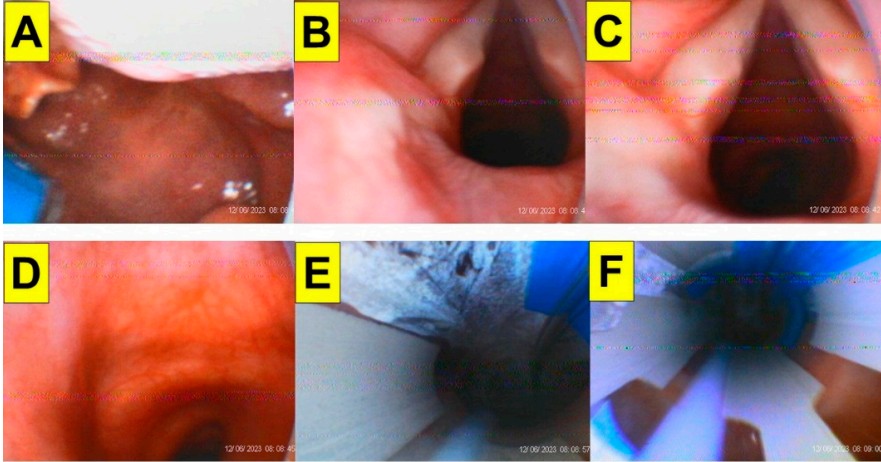

**Figure 8.** (Case 2 in Table 1) Serial images recorded during styletubation. (**A–F**): Views acquired by styletubation show various signposts, e.g., epiglottis, vocal cords, trachea, and the EMT-ET tube. (See Video S2 in the Supplementary Materials).

## 2. Discussion

In this case report, we present 15 cases in which both the tracheal intubation and verification of the proper position of an EMG-ET tube were conducted by a combined styletubation technique with videolaryngoscopy in patients undergoing a thyroidectomy with IONM.

The application of IONM during a thyroidectomy, to visualize and identify the RLN, is crucial and beneficial to lessen complications of nerve injury and reduce the sequelae of postoperative vocal cord paralysis. Such an intraoperative neuromonitoring device and technique to enhance the identification of the RLN by providing functional dynamics of an evoked EMG requires a proper position and adequate surface contact between the surface electrodes over the endotracheal tube and vocal cords (featured by the conductive silver ink electrodes and a cross-band to guide placement). Usually, the proper placement and verification of the position of the EMG-ET tube required optic devices (e.g., VL and FOB) to be achieved (Table 2). It was also our purpose to simultaneously apply VL with styletubation to verify the proper position of the EMG-ET tube after the placement of the tube (Figure 1).

**Table 2.** Comparison of effectiveness of various intubating modalities for tracheal intubation in patients undergoing thyroidectomy.

| | DL | VL | FOB | Styletubation |
|---|---|---|---|---|
| Availability and affordability | +++ | ++ | + | ++ |
| Difficult intubation with IONM tube | 6.5% [44] | - | - | 2.7% [44]<br>0% (this article) |
| First-pass success rate | 75% [20]<br>96.4% [44] | 97.5% [20] | - | 90% [20]<br>99% [44]<br>100% (this article) |
| Time to intubate | 68.8 s [20] | 29.8 s [20] | - | 42.4 s [20]<br>4–15 s (this article) |
| Initial successful placement without tube depth adjustment | 94.3% [36]<br>96.4% and<br>87.3% [37]<br>33.5% [42] | 67% [40]<br>97.5% [41]<br>100% [49]<br>44% [42] | 100% [38] | 100% [50]<br>100% (this article) |

Currently, airway operators used to apply either conventional laryngoscopy (DL) or videolaryngoscopy (VL) for routine tracheal intubation. The advantages and disadvantages of DL, VL, and FOB for placement and verification of an EMG-ET tube for IONM have been occasionally compared in the literature (Table 2). In our medical center, we have universally applied the styletubation technique for routine tracheal intubation in the past 7 years [46–48]. In particular, we appreciate the superiority of the styletubation technique in the placement of special endotracheal tubes, e.g., double-lumen endobronchial tubes and EMG-ET tubes. When styletubation was applied for tracheal intubation with an EMG-ET tube, the effectiveness and accuracy were reflected by its time to intubate, first-pass success rate, and airway operator's subjective satisfaction (Tables 1 and 2). It should be mentioned that the overall incidence of DI incidence in patients undergoing a thyroidectomy was reported to be about 4.4%, and that was significantly lower with the VS technique (2.7%) than with DL (6.5%) [44]. Our clinical experience of styletubation to place the EMG-ET tube is outstanding (i.e., first-pass success rate and time to intubate) as shown in Table 1. It is worthy to mention that the presence of any giant thyroid mass over the neck definitely upgraded the intubation difficulty scale score and lengthened the time to intubation by either VL or styletubation. Difficult airway management (in particular, tracheal intubation) may be encountered in patients with thyroid anatomical abnormalities. Therefore, adequate neuromuscular relaxation by neuromuscular blocking agents (NMBAs)

before performing tracheal intubation is deemed necessary and helpful. While being good for tracheal intubation, the profound muscular relaxing effects of the NMBAs definitely would affect the effectiveness and accuracy of the acquisition and interpretation of the EMG signals of the IONM during a thyroidectomy. Most of the time, the residual effects of the first intubating dose of the NMBAs immediately after induction of anesthesia has not yet weaned off before the surgeon starts dissection and requires IONM signals. Therefore, it is reasonable to optimize the types, dosages, and timing of the NMBAs for the intubating doses [51,52] or timely using the reversal/chelating agents (neostigmine or sugammadex) before properly acquiring IONM signals [53–57]. It is worth mentioning that the application of the combined styletubation with videolaryngoscopy technique is in particular technically advantageous so that no profoundly deep neuromuscular blockade is required for tracheal intubation. Therefore, it is easier to place the EMG-ET tube without using full-dose NMBAs for tracheal intubation while timely regaining the muscular functions for IONM. Several dosing protocols of the NMBAs include, e.g., a reduction in the full dose of rocuronium to half or one-third of the recommended doses, with supplementary doses of succinylcholine, helping to gain a good V1 signal in a timely fashion. If the neuromuscular blockade did not wean off in time and interfered with the EMG recording, sugammadex was used to antagonize such a blockade by rocuronium. Commonly, a combined small dose of rocuronium and full-dose succinylcholine can provide an adequate neuromuscular blockade for tracheal intubation while preserving the effectiveness and accuracy of subsequent IONM. Other short-acting NMBAs, e.g., mivacurium, also serve such a purpose for a thyroidectomy with IONM. For a thyroidectomy, the patient is placed in a supine position with a shoulder roll placed behind the patient's upper back so the torso would be elevated, exposing and hyperextending the neck. Such positioning of the patient may cause malposition of the intubated endotracheal tube, e.g., migration outward to a certain length. Therefore, it has been suggested first to place the patient in a surgery-fit position and then perform the tracheal intubation, if the pre-operative airway evaluation did not predict a difficult airway scenario (Figures 7 and 8). After the tracheal intubation is accomplished, the airway operator needs to check and verify that the length and position of the EMG-ET tube are correct and proper. This end can be achieved by using DL, VL [37], or FOB [36] to obtain the images of the position of the EMG-ET tube. Our practice with the combined styletubation and VL technique provided a smooth tracheal intubation (by the former) and verified the proper position of the EMG-ET tube (by the latter). It was proposed that a smaller ET tube size for a thyroidectomy may potentially have better postoperative vocal outcomes, incidence of laryngeal trauma as assessed by laryngoscopy, or pain scores [58]. However, the adequate vocal cords contact surface area and proper depth of the EMG-ET tube for IONM determined the quality of the acquired signals.

The paradigm shift of VL (versus DL) for tracheal intubation can never be overemphasized and has recently been debated for its standard-of-care position [59]. In the literature, VL has demonstrated its advantages in various airway management scenarios, including a simulated difficult airway [60], obesity [61], applying a double-lumen endobronchial tube [62], and during the COVID-19 pandemic [63,64]. Similarly, a new paradigm role of styletubation (using a video-assisted intubating stylet for tracheal intubation) has been proposed [46–48]. Styletubation has also been applied in various difficult airway scenarios, including limited cervical mobility, obesity, head–neck tumor or fibrosis, double-lumen endobronchial tube placement, rapid-sequence induction, and the COVID-19 pandemic [65].

## 3. Conclusions

In this case series clinical report, we demonstrate 15 cases receiving an EMG-ET tube for IONM during a thyroidectomy. Combined styletubation with VL for such a clinical purpose provides a clinical effectiveness and accuracy to ensure the placement of an EMG-ET tube is correct and smooth.

**Supplementary Materials:** The following supporting information can be downloaded at: https://www.mdpi.com/article/10.3390/anesthres1010003/s1.

**Author Contributions:** Conceptualization, H.-S.P., T.C., H.-N.L., J.Z.Q. and A.S.; methodology, H.-S.P. and H.-N.L.; validation H.-S.P., T.C. and H.-N.L.; formal analysis, H.-S.P. and H.-N.L.; investigation, H.-S.P. and H.-N.L.; resources, H.-N.L.; data curation, H.-S.P. and H.-N.L.; writing—original draft preparation, H.-S.P., T.C. and H.-N.L.; writing—review and editing, J.Z.Q. and A.S.; visualization, H.-N.L.; supervision, H.-N.L., J.Z.Q. and A.S.; project administration, H.-N.L.; funding acquisition, H.-N.L. All authors have read and agreed to the published version of the manuscript.

**Funding:** This research received no external funding.

**Institutional Review Board Statement:** This study was conducted according to the guidelines of the Declaration of Helsinki and was approved by REC, Hualien Tzuchi Hospital (approved letter number: CR112-08).

**Informed Consent Statement:** Written informed consent was obtained from all patients or their legally authorized representatives.

**Data Availability Statement:** Not applicable.

**Acknowledgments:** This work was supported by Hualien Tzuchi Hospital.

**Conflicts of Interest:** The authors declare no conflict of interest.

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
