# Peer review of "Combined Styletubation with Videolaryngoscopy for Tracheal Intubation in Patients Undergoing Thyroidectomy with Intraoperative Neuromonitoring"

_2813-5806, doi:10.3390/anesthres1010003_

Round 1
Reviewer 1 Report
The present manuscript is an interesting well-written case report series regarding airway management in thyroidectomy in combination with intra-operative neuro-monitoring.
In this manuscript, the currently common intubation procedures are described and the version of a combination of a styletubation with videolaryngoscopy presented is adequately classified in this context: It is another option of airway management without the authors significantly elevating it above the other options:
Only the “time to intubate” shown in Table 2 from the study by Liu et al. does not really seem to fit in terms of comparability of the degree of difficulty of intubation. In the study by Liu et al., all intubation processes (convention intubation, videolaryngoscopic intubation and styletubation) were significantly longer than in the present manuscript. In the manuscript by Liu et al., however, "thyroid tumor patients with a difficult airway" were examined. In this regard, I ask for comparison with the patients who are certainly much easier to intubate in this manuscript – please explain and discuss.
Minor aspects:
Page 4, line 120: sevoflurane or desflurane
Page 4, line 120: a) why do you use rocuronium instead of mivacurium? Please discuss.
b) please add in Table 1 the dosage of rocuronium (mg/kg) for each patient
c) What was the value of the TOF for each patient at the time of the incision
d) did you use suggamadex?
e) Please address points a) to d) in the discussion (page 12)
Page 2, line 66: intra-operative instead of intro-operative
Author Response
Response to the Reviewer 1
The present manuscript is an interesting well-written case report series regarding airway management in thyroidectomy in combination with intra-operative neuro-monitoring.
In this manuscript, the currently common intubation procedures are described and the version of a combination of a styletubation with videolaryngoscopy presented is adequately classified in this context: It is another option of airway management without the authors significantly elevating it above the other options:
Comment-1: Only the “time to intubate” shown in Table 2 from the study by Liu et al. does not really seem to fit in terms of comparability of the degree of difficulty of intubation. In the study by Liu et al., all intubation processes (convention intubation, videolaryngoscopic intubation and styletubation) were significantly longer than in the present manuscript. In the manuscript by Liu et al., however, "thyroid tumor patients with a difficult airway" were examined. In this regard, I ask for comparison with the patients who are certainly much easier to intubate in this manuscript – please explain and discuss.
Response-1: Thanks for your great question. Actually, in the Table 2, we listed not only the “time to intubate”, but also “first-pass success rate”. The authors acknowledged the limitation of the scanty literature regarding the “comparative studies” among DL, VL, and VS in patients with “giant” or “huge” thyroid mass. It is absolutely correct that the size and location of thyroid tumor mass would upgrade the level of difficulty. Unfortunately, even in the reference-20, ADS (or IDS) values and DA predictors in the Table-2 did not clearly reflect the association with thyroid mass size (e.g., in Figure 1). In addition, the study device of the reference-20 was not EMG-ET tube for IONM. Also, due to the ethics issue, one would expect the difficult to conduct such comparative study or even RCT in such airway management topic.
“It is worthy to mention that the presence of any giant thyroid mass over the neck definitely upgraded the intubation difficulty scale score and lengthened the time to intubation by either VL or styletubation.” has been added into the revised text.
Minor aspects:
Comment-2: Page 4, line 120: sevoflurane or desflurane
Response-2: Has been corrected as “sevoflurane or desflurane”. Thanks.
Comment-3: Page 4, line 120: why do you use rocuronium instead of mivacurium? Please discuss.
Response-3: Great comment. Our prior clinical use experience with mivacurium was 20 years ago. With the plasma cholinesterase metabolism features, the 20-min duration seems to be feasible for short duration of surgery e.g., LMS, and IONM-thyroidectomy. Unfortunately, due to various reasons (including business issue from the pharmaceutics), mivacurium is not available here locally.
“Other short-acting NMBAs, e.g., mivacurium, also serves such a purpose for thyroidectomy with IONM.” has been added into the revised text.
Comment-4: please add in Table 1 the dosage of rocuronium (mg/kg) for each patient
Response-4: Thanks for your point. Actually, in the previous version of the Table-1, we have already put the induction dose of ROC for each patient presented as “mg/kg”.
Comment-5: What was the value of the TOF for each patient at the time of the incision.
Response-5: Thanks for this great question. We did not record the TOF ratio value in each patient when the surgeon made the skin incision. This is because this manuscript is a case series report, not designed as a prospective clinical study. Therefore, the pre-planned collection of various data was not possible. However, the residual neuromuscular blockade after optimally performed tracheal intubation was continuously monitored. Meanwhile, the time span between complete induction of anesthesia and skin incision varied and was surgeon-dependent. Usually, it took about 20 min to 30 min (mainly time for the setting-up). When that happened, the value of TOF ratio after the first intubating dose of rocuronium (or rocuronium plus succinylcholine) went back. The prime objective is to allow the surgeon properly to record V1 signal (vagus nerve stimulation) and R1 signal (RLN). In comparison to use of full dose of ROC (0.6 mg/kg), half-dose (0.3 mg/kg) or one third-dose (0.2 mg/kg) usually could help attaining good signals. If not, sugammadex is the choice to reverse the residual effects of ROC. In conclusion, either reducing dose of ROC (or in combination with SCh) or supplementary chelating dose of sugammadex helps to gain a good V1 signal in a timely fashion.
“Several dosing protocols of the NMBAs include, e.g., reduction in the full-dose of rocuronium to half or one-third of the recommended doses, with supplementary doses of succinylcholine, help to gain a good V1 signal in a timely fashion.” has been added into the revised text.
Comment-6: did you use suggamadex?
Response-6: Yes, we did use sugammadex when necessary (e.g., failure to record V1 signal). However, because the cost of sugammadex goes to patients under our universal health plan (national health insurance program). Therefore, it is not routinely administered. In that case, the surgeon just waited for a little longer. This time delay may be improved by given a smaller dose of ROC (e.g., 1/3 dosage) plus SCh for tracheal intubation.
“If the neuromuscular blockade did not wean off in time and interfered EMG recording, sugammadex was used to antagonize such blockade by rocuronium.” has been added into the revised text.
Comment-7: Please address points a) to d) in the discussion (page 12)
Response-7: The responses have been reflected in the revised text accordingly. Thanks.
Comment-8: Comments on the Quality of English Language. Page 2, line 66: intra-operative instead of intro-operative
Response-8: The “intra-operative” has been corrected in the text. Thanks for your correction.

Reviewer 2 Report
I must say, this case series report on tracheal intubation using the evoked electromyography-endotracheal tube (EMG-ET tube) for continuous intra-operative recurrent laryngeal nerve monitoring (IONM) in thyroid surgery patients is truly impressive. The researchers have successfully demonstrated the current state of the art in this field, and their findings are undoubtedly groundbreaking.
The use of both direct laryngoscopy (DL) and videolaryngoscopy (VL) for routine tracheal intubation of the EMG-ET tube is fascinating and highlights the versatility of this technique. However, what truly caught my attention is the innovative "styletubation" method, which employs a video-assisted intubating stylet (VS) for smooth and swift intubation. This approach seems to be a game-changer, ensuring efficient and precise placement of the EMG-ET tube.
Moreover, the researchers' decision to combine styletubation with VL appears to be a stroke of brilliance, as it further enhances the accuracy and reliability of the final tube placement. Such a combination undoubtedly elevates the overall quality and effectiveness of intubating EMG-ET tube for IONM.
However, I believe that the bibliography should be supplemented with, among others, the following publications :
doi: 10.1177/0145561320974829.
doi: 10.5603/DEMJ.a2022.0004
doi: 10.1001/jamaoto.2015.1198.
doi: 10.2298/vsp0905377k.
doi: 10.5603/DEMJ.a2021.0012
doi: 10.21037/jtd-22-451.
doi: 10.3390/children9111774.
doi: 10.5603/DEMJ.a2020.0023
doi: 10.1111/anae.15865.
doi: 10.1016/j.ijscr.2016.06.018.
Author Response
Response to Reviewer 2
I must say, this case series report on tracheal intubation using the evoked electromyography-endotracheal tube (EMG-ET tube) for continuous intra-operative recurrent laryngeal nerve monitoring (IONM) in thyroid surgery patients is truly impressive. The researchers have successfully demonstrated the current state of the art in this field, and their findings are undoubtedly groundbreaking.
The use of both direct laryngoscopy (DL) and videolaryngoscopy (VL) for routine tracheal intubation of the EMG-ET tube is fascinating and highlights the versatility of this technique. However, what truly caught my attention is the innovative "styletubation" method, which employs a video-assisted intubating stylet (VS) for smooth and swift intubation. This approach seems to be a game-changer, ensuring efficient and precise placement of the EMG-ET tube.
Moreover, the researchers' decision to combine styletubation with VL appears to be a stroke of brilliance, as it further enhances the accuracy and reliability of the final tube placement. Such a combination undoubtedly elevates the overall quality and effectiveness of intubating EMG-ET tube for IONM.
However, I believe that the bibliography should be supplemented with, among others, the following publications :
- doi: 10.1177/0145561320974829.
- doi: 10.5603/DEMJ.a2022.0004
- doi: 10.1001/jamaoto.2015.1198.
- doi: 10.2298/vsp0905377k.
- doi: 10.5603/DEMJ.a2021.0012
- doi: 10.21037/jtd-22-451.
- doi: 10.3390/children9111774.
- doi: 10.5603/DEMJ.a2020.0023
- doi: 10.1111/anae.15865.
- doi: 10.1016/j.ijscr.2016.06.018.
Response: The authors sincerely appreciate the reviewer’s excellent comments with great vision and professionalism. A truly airway management expert. We take your compliment as a driving force and motivation to go forward. Many thanks.
As suggested, we have revised the text and added the 9 recommended references (among them, the “doi: 10.1177/0145561320974829” Is the same as [42])
In addition, we added two more references ([59],[65]) to draw attention on the updated issues of the VL and VS.
- [15]. Kalezić, N.; Milosavljević, R.; Paunović, I.; Zivaljević, V.; Diklić, A.; Matić, D.; Ivanović, B.; Nesković, V. The incidence of difficult intubation in 2000 patients undergoing thyroid surgery--a single center expirience. Vojnosanit Pregl. 2009, 66, 377-382. doi: 10.2298/vsp0905377k.
- [25]. Watt, S.; Kalpan, J.; Kolli, V. Case report of the use of videolaryngoscopy in thyroid goiter masses: An airway challenge. J. Surg. Case Rep. 2016, 27, 119-121. doi: 10.1016/j.ijscr.2016.06.018.
- [43]. Kriege, M.; Hilt, J.A.; Dette, F.; Wittenmeier, E.; Meuser, R.; Staubitz, J.I.; Musholt, T.J. Impact of direct laryngoscopy vs. videolaryngoscopy on signal quality of recurrent laryngeal nerve monitoring in thyroid surgery: a randomised parallel group trial. Anaesthesia 2023, 78, 55-63. doi: 10.1111/anae.15865.
- [58]. Mehanna, R.; Hennessy, A.; Mannion, S.; O'Leary, G.; Sheahan, P. Effect of endotracheal tube size on vocal outcomes after thyroidectomy: A randomized clinical trial. JAMA Otolaryngol. Head Neck Surg. 2015, 141, 690-695. doi: 10.1001/jamaoto.2015.1198.
- [59]. Aziz, M.F.; Berkow, L. Pro-con debate: Videolaryngoscopy should be standard of care for tracheal intubation. Anesth Analg. 2023, 136, 683-688. doi: 10.1213/ANE.0000000000006252.
- [60]. Gungorer, B.; Findik, M.; Kayipmaz, A.E. USB-endoscope laryngoscope is as effective as video laryngoscope in difficult intubation. Disaster Emerg. Med. J. 2021, 6, 75–79. DOI: 10.5603/DEMJ.a2021.0012.
- [61]. Evrin, T.; Szarpak, L.; Katipoglu, B.; Mishyna, N.; Kockan, B.S.; Ruetzler, K.; Schläpfer, M. Video-assisted versus macintosh direct laryngoscopy for intubation of obese patients: a meta-analysis of randomized controlled trials. Disaster Emerg. Med. J. 2022, 7, 30-40. doi: 10.5603/DEMJ.a2022.0004.
- [62]. Palaczynski, P.; Misiolek, H.; Bialka, S.; Owczarek, A.J.; Gola, W.; Szarpak, L.; Smereka, J. A randomized comparison between the VivaSight double-lumen tube and standard double-lumen tube intubation in thoracic surgery patients. Thorac. Dis. 2022, 14, 3903-3914. doi: 10.21037/jtd-22-451.
- [63]. Wieczorek, P.; Szarpak, L.; Dabrowska, A.; Pruc, M.; Navolokina, A.; Raczynski, A.; Smereka, J. A comparison of the bébé VieScope™ and direct laryngoscope for use while wearing PPE-AGP: A randomized crossover simulation trial. Children (Basel). 2022, 9(11):1774. doi: 10.3390/children9111774.
- [64] Ludwin K.; Bialka, S.; Czyzewski, L.; Smereka, J.; Marek Dabrowski, M.; Dabrowska, A.; Ladny, J.R.; Ruetzler, K.; Szarpak, L. Video laryngoscopy for endotracheal intubation of adult patients with suspected/ confirmed COVID-19. A systematic review and meta-analysis of randomized controlled trials. Disaster Emerg. Med. J. 2020, 5, 85-97. DOI: 10.5603/DEMJ.a2020.0023
- [65] Tsai, P.B.; Luk, H.-N. Sheet Barrier and Intubating Stylet. Encyclopedia 2021, 1, 1058-1075. https://doi.org/10.3390/encyclopedia1040081
- [42]. Casano, K.; Cannon, C.R.; Didlake, R.; Replogle, W.R.; Cannon, R. Use of GlideScope in patients undergoing NIM thyroidectomy. Ear Nose Throat J. 2022, 101, 650-653. doi: 10.1177/0145561320974829.
